# The Prevalence of Undiagnosed *Salmonella enterica* Serovar *Typhi* in Healthy School-Aged Children in Osun State, Nigeria

**DOI:** 10.3390/pathogens12040594

**Published:** 2023-04-14

**Authors:** Jessica N. Uwanibe, Tolulope A. Kayode, Paul E. Oluniyi, Kazeem Akano, Idowu B. Olawoye, Chinedu A. Ugwu, Christian T. Happi, Onikepe A. Folarin

**Affiliations:** 1African Center of Excellence for Genomics of Infectious Diseases (ACEGID), Redeemer’s University, Ede 232103, Osun State, Nigeria; uwanibej@run.edu.ng (J.N.U.); kayodet@run.edu.ng (T.A.K.);; 2Department of Biological Sciences, College of Natural Sciences, Redeemer’s University, Oshogbo 232102, Osun State, Nigeria

**Keywords:** *Salmonella Typhi*, typhoid fever, ELISA, lipopolysaccharide, asymptomatic, next-generation sequencing

## Abstract

Typhoid fever remains a significant public health concern due to cases of mis-/overdiagnosis. Asymptomatic carriers play a role in the transmission and persistence of typhoid fever, especially among children, where limited data exist in Nigeria and other endemic countries. We aim to elucidate the burden of typhoid fever among healthy school-aged children using the best surveillance tool(s). In a semi-urban/urban state (Osun), 120 healthy school-aged children under 15 years were enrolled. Whole blood and fecal samples were obtained from consenting children. ELISA targeting the antigen lipopolysaccharide (LPS) and anti-LPS antibodies of *Salmonella Typhi*, culture, polymerase chain reaction (PCR), and next-generation sequencing (NGS) were used to analyze the samples. At least one of the immunological markers was detected in 65.8% of children, with 40.8%, 37.5%, and 39% of children testing positive for IgM, IgG, and antigen, respectively. Culture, PCR, and NGS assays did not detect the presence of *Salmonella Typhi* in the isolates. This study demonstrates a high seroprevalence of *Salmonella Typhi* in these healthy children but no carriage, indicating the inability to sustain transmission. We also demonstrate that using a single technique is insufficient for typhoid fever surveillance in healthy children living in endemic areas.

## 1. Introduction

Typhoid fever remains a significant disease, especially in low and middle-income countries (LMICs), due to poor/lack of access to potable water, unhygienic practices, and poverty [1]. The disease is caused by *Salmonella Typhi*, with a global estimate of 21.6 million new infections annually, 210,000 typhoid fever-related deaths, and 5.4 million cases of paratyphoid fever [2,3]. Over 70% of international cases are in Asian (Bangladesh, India, and Pakistan) and African (383 per 100,000 persons per year) countries [4,5,6,7]. Although the global prevalence of typhoid fever has reduced in recent years by 43% [8], the case-fatality rate remains high (10–30%) due to a lack of effective treatment [3]. To date, only a limited amount of non-recent data exists on the actual burden of typhoid fever in several LMICs [6]. In Nigeria, Africa’s most populous country, the available information on the burden of typhoid fever is scanty [9]. Previous reports in Africa have shown that children bear the highest burden of typhoid disease [10]. Asymptomatic carriers play a role in typhoid fever burden in endemic areas. Such cases go undetected and may serve as carriers and reservoirs of the pathogen, especially in the endemic region [11]. The screening of individuals who are silent carriers will be helpful in controlling the spread of Salmonella infection among the public. It is estimated that two to five percent of acute typhoid cases resolve into chronic carriers, thereby facilitating typhoid fever transmission and persistence in the human population [12,13]. This continuous transmission within the human population will hinder achieving the sustainable development goal (SDG 3) of healthy lives and promoting well-being for all. There is a need to elucidate the actual burden of typhoid fever in order to achieve disease elimination and eradication. This can be accomplished by actively surveilling symptomatic and asymptomatic children who serve as reservoirs and carriers of *Salmonella Typhi* [14,15]. This will routinely identify *Salmonella Typhi* within suspected carriers, in this case, healthy children who maintain the infection with no symptoms in this environment and may transmit it to other children. Inadequate and lack of access to proper diagnosis in Nigeria and other endemic countries leads to wrong diagnoses of typhoid fever and makes it challenging to elucidate the actual burden of the disease. Over the years, the Widal test, an agglutination-based assay, has been the most deployed test for typhoid fever diagnosis [16,17,18]. However, the Widal test is cross-reactive with other *Salmonella enterica* serovar species and bloodstream pathogens, being unable to distinguish between active and prior infections or natural and vaccine-induced immunity [19]. The gold standard for typhoid fever diagnosis is the culture method by isolating *Salmonella Typhi* from blood or bone marrow samples [20]. However, blood cultures, the gold standard for detection, are often unsuccessful due to the indiscriminate use of antibiotics, readily available as over-the-counter drugs, inhibiting bacteria growth. In addition, the required laboratory equipment and expertise are not readily available [3,21].

There is a need to utilize other diagnostic techniques that will be able to circumvent the limitations associated with the current Widal test. The enzyme-linked immunosorbent assay (ELISA) approach provides a better alternative to the Widal test. It is more sensitive and specific for *Salmonella Typhi* detection, targeting specific antigen protein in the organism. It can also distinguish between active and prior infections [22]. It is noteworthy that the use of ELISA for typhoid fever prevalence is underutilized [23]. Molecular techniques, such as polymerase chain reaction (PCR) and next-generation sequencing, provide a better alternative to bacteria culture because it detects the bacteria nucleic materials and thus will mitigate the concern of antibiotic interference.

To elucidate the burden of asymptomatic cases of typhoid fever in healthy school-aged children in Osun State, Nigeria, and to determine the best-employed techniques for surveillance, we used anti-lipopolysaccharide (anti-LPS) immunoglobulin G (IgG), anti-LPS Immunoglobulin M (IgM), and antigen LPS *Salmonella Typhi* specific ELISA, culture method, polymerase chain reaction, and next generation sequencing approaches.

To our knowledge, the application of these techniques for asymptomatic cases of typhoid fever is the first of its kind in Nigeria.

## 2. Methodology

### 2.1. Study Design

This study was conducted between December 2019 and February 2020. Healthy school-aged children visiting the LAUTECH teaching hospital, Osun State, for blood grouping and genotyping, and primary school children who volunteered to be part of the study were enrolled. The sample size was calculated based on a previous study [9] with a power of 93% and an error margin of 5%.

### 2.2. Ethical Approval

Ethical clearance for this study was obtained from the research ethics committee of the Ladoke Akintola University of Technology Teaching Hospital (LAUTECH) (LTH/EC/2019/09/431) and Ministry of Health (OSHREC/PRS/569T/164) Oshogbo, Osun State, Nigeria.

### 2.3. Inclusion Criteria

School-aged children below the age of 15 years who showed no symptoms of any febrile illness and who were willing to participate in the study with written informed consent from their parents/legal guardians.

### 2.4. Exclusion Criteria

Children who showed signs of febrile illness, or children or parents unwilling to consent to participate in the study.

### 2.5. Sample Collection and Processing

Five milliliters (5 mL) of venous blood were obtained from each of the 120 enrolled children and transferred to EDTA bottles. Sixty-seven (67) fecal samples were obtained from some of the enrolled children following written consent from parents/guardians and assent from children between the ages of 8–15 years. Whole blood (W.B.) and fecal samples were transported in a cold chain to the African Centre of Excellence for Genomics of Infectious Diseases (ACEGID) and Biological Sciences Laboratories, respectively, at Redeemer’s University Ede, Osun State, for processing. We obtained non-haemolysed plasma samples by centrifugation (at 1500× *g* for 10 min) and stored them at −20 °C before ELISA analysis.

### 2.6. Antigen and Antibodies Detection Using the Enzyme-Linked Immunosorbent Assay (ELISA)

ELISA for detecting *Salmonella Typhi* IgM, IgG anti-LPS antibodies, and *Salmonella Typhi* LPS antigen was carried out according to the manufacturer’s instructions using the Human salmonella typhoid kit from Melsin Medical Co. Limited (Cat. No. EKHU-2050, EKHU-0553 and EKHU-0883). The kit targets the LPS O-group antigen and antibody in the assayed samples. Positive and negative controls provided by the manufacturers served as standards for calling positive or negative results. The absorbance of each well was read on a spectrophotometer O.D. at 450 nm, as described by Felgner and others [24].

### 2.7. Bacterial Culture for Isolation of the Salmonella Typhi

Fecal samples from 67 healthy children were cultured using selenite-F broth (Oxoid) for 16 h at 37 °C. The broth was subcultured on *Salmonella* and *Shigella* agar at the same temperature for 18 h. An analytical profile index (20E API) test (bioMerieux) was used for isolate identification.

### 2.8. Molecular Detection of Salmonella Typhi Using Polymerase Chain Reaction (PCR)

DNA was extracted from whole blood using the Qiagen Dneasy Blood and Tissue kit (Qiagen, Hilden, Germany). The bacterial DNA from the fecal samples was released and purified using the Zymo Quick-DNA™ Miniprep kit (Zymo, Irvine, CA USA) for fecal and soil samples according to the manufacturer’s protocol. DNA regions of H (Flagella) antigen and O group genes for *Salmonella Typhi* and *Salmonella Paratyphi A* were amplified. The multiplex PCR was optimized to a final volume of 25 µL using PCR ready-to-go beads in 1.5 mM MgCl2 and 0.8 µM each of forward and reverse primers (Table 1) and 5 µL of the extracted DNA template. The PCR program used for amplification consisted of 2 min at 95 °C initial denaturation, followed by 35 cycles of 30 s at 95 °C of denaturing temperature, 15 s at 51 °C of annealing temperature, and 30 s at 72 °C of extension temperature, then 5 min at 72 °C of final extension temperature. This protocol was optimized according to the method of Levy et al. [25].

### 2.9. Whole Genome Sequencing of Salmonella Typhi Using the Next Generation Sequencing Technique

The concentration of extracted DNA samples was determined using 1X dsDNA High Sensitivity assay (Invitrogen, Waltham MA, USA) in a Qubit fluorometer (ThermoFisher Scientific, Waltham MA, USA). Sequencing libraries were prepared using the Illumina DNA preparation kit (Illumina, San Diego CA, USA). Library preparation protocol was adopted from the CDC PulseNet Nextera DNA Flex Standard operating protocol [26] and sequenced using the Illumina Miseq platform at the African Center of Excellence for Genomics of Infectious Diseases (ACEGID), Redeemer’s University, Nigeria.

### 2.10. Data Analysis

The ELISA data generated from the microplate reader were analyzed using Microsoft Excel for Mac (version 16.50). Data were double-entered to reduce data entry errors and later merged. Delta O.D. (Optical density) (O.D. at 630 nm subtracted from O.D. at 450 nm) for each sample was blanked by subtracting the O.D. value of the blank. According to the manufacturer’s instructions, a cutoff value was calculated by adding 0.15 to the O.D. of the negative control. Samples with an O.D. less than or equal to the cutoff were considered seronegative, while samples with higher O.D. values than the cutoff were deemed seropositive for IgG, IgM, or antigen. Statistical comparisons were made using SPSS version 25.0, Epi-info version 7, and GraphPad Prism version 8.4.2.

Discrete variables were compared by chi-square with Yates’ correction or Fisher’s exact test. Normally distributed, continuous variables were compared by Student’s t-test and analysis of variance (ANOVA). Post hoc comparisons were made using Turkey’s honest significant test. Non-parametric variables were compared using Mann–Whitney U tests or Kruskal–Wallis tests. Pearson or Spearman’s rank correlation coefficient evaluated the relationship between two continuous variables (such as O.D.s for IgM versus IgG, antigen versus IgM or IgG). All tests of significance were two-tailed, and *p*-values < 0.05 were taken to indicate significant differences.

### 2.11. Quality Control and Taxonomic Classification

To improve the quality of the FASTQ reads, they were processed with Trimmomatic [27] to filter low-quality bases (<20 Phred score), short reads (<50 bp), and Illumina adapters. The resulting paired-end FASTQ files were analyzed with Kraken2 [28] extensive database. Individual reports from Kraken2 output were merged in a single CSV report file using the aggregate_metagenomics_report workflow of viral-ngs v2.1.18.0 [29], stratifying the report to display taxonomy levels of the top three genera of bacteria present in all samples.

## 3. Results

### 3.1. Demography of Study Participants

We enrolled 120 school-aged children, 71 (59.2%) male and 49 (40.8%) females. The mean age of all participants was 6.1 ± 2.9 years (range 1–14) (Table 2).

### 3.2. High Incidence of Salmonella Typhi Antibodies and Antigen in Non-Symptomatic Healthy Children

We investigated *Salmonella Typhi* infection in the cohort of healthy children using semi-quantitative antigen and antibody ELISA. It targets *Salmonella Typhi* LPS and IgM, IgG anti-lipopolysaccharides (anti-LPS) antibodies (Melsin Human salmonella typhoid ELISA). A total of 79 (65.8%) children were positive for at least one of the immunological markers, while 41 (34.2%) children were negative for all the markers (IgG, IgM, and Antigen). Both anti-LPS IgG and IgM were detected in 69 (57.5%) samples, and all three markers (anti-LPS IgG, IgM, and LPS antigen) were seen in 17 (14%) samples (Figure 1).

A high proportion of children elicited anti-LPS antibody responses, and both the prevalence of anti-LPS IgM (49 (40.8%) and anti-LPS IgG (45(37.5) were similar in the children (Figure 1). The mean O.D. and standard deviation value for positive IgM and IgG assays were 0.2554 ± 0.0797 (ranging from 0.1640 to 0.4805) and 0.3283 ± 0.1219 (ranging from 0.1740 to 0.6085), respectively (Figure 2). A similar proportion was observed between males (26.6%) and females (46.9%) for the anti-LPS IgM responses. In comparison, 36.6% of males and 38.8% of females were positive for anti-LPS IgG. These differences were not statistically significant, with *p* = 0.35 (IgM) and *p* = 0.96 (IgG). Comparing children five years and under with children over five years: 38.6% of children ≤ 5 years and 42.9% of children > 5 years tested positive for IgM antibodies, while 29.8% of children ≤ 5 years and 44.4% of children > 5 years were positive for IgG antibodies.

Our study detected LPS antigen in 47 (39%) children enrolled in the study. While most of the children had antigens and one or both anti-LPS antibodies (37), only ten children tested positive for LPS antigens alone (Figure 1). The mean O.D. and standard deviation value for the positive antigen assay were 0.3836 ± 0.1887 (ranging from 0.1535 to 1.1035) (Figure 3). Positive antigens were similar in males and females (29 of 71 (40.8%) versus 18 of 49 (36.7%) *p* = 0.79) and in those aged ≤5 years and >5 years (19 of 59 (32.2%) versus 28 of 63 (44.4%); respectively. *p* = 0.29).

Distribution of all confirmed immunological markers for *Salmonella Typhi* based on age was highest in children above five years. Based on the percentage of O.D., anti-LPS IgG (62.2%), anti-LPS IgM (55.1%), and *Salmonella Typhi* LPS antigen (59.6%) were observed (Figure 2 and Figure 3, respectively).

### 3.3. Bacteria Culture in Detecting Salmonella Typhi in Healthy Children

We collected stool samples from the children in our cohort and tested them for bacteria growth. All cultured fecal samples (67) were negative for *Salmonella Typhi*. However, twenty-three (23) isolates showed positive phenotypic (black-centered colonies with transparent boundaries) results on Salmonella/Shigella agar (SSA). Further confirmation using a 20E API kit (bioMerieux Marcy-l’Étoile, France) showed that all were negative for *Salmonella Typhi* but were positive for *Citrobacter* species (Table 3).

### 3.4. The Molecular Approach to Detecting Salmonella Typhi in Healthy Children

Consequently, we selected antigen-positive samples for further confirmation using PCR targeting the *Salmonella Typhi* and *Salmonella Paratyphi A* H (flagellin) antigen and O (somatic) group genes. All 47 antigen-positive samples showed no amplification for *H*(*d*) *antigen* and O group genes for *Salmonella Typhi*. Positive amplification for *Salmonella Paratyphi A* was seen in six samples with high O.D. values ranging from 0.6022–1.1035 Table 4.

Five (5) samples with high O.D. (ranging from 0.4080–1.1035) values and four (4) antigen LPS negative samples were randomly selected for metagenomics sequencing on their corresponding fecal samples to establish *Salmonella Typhi* carriage. Metagenomics sequencing showed no *Salmonella Typhi* nor other Salmonella species reads in the fecal samples. However, other known gut microbiotas were detected in all samples except for one due to low sequencing quality. Some of these microbiotas are known to be pathogenic to humans but, to the best of our knowledge, are not reported in cross-reactivity with *Salmonella Typhi* ELISA (Figure 4).

## 4. Discussion

This study elucidated the prevalence and carriage of *Salmonella Typhi* in apparently healthy children in Osun State, Nigeria, by employing ELISA, culture, PCR, and next generation sequencing. Children, especially those in the endemic area, are more susceptible to typhoid fever [30]. Our results show a high prevalence of *S. Typhi* in these healthy children (Figure 1). Our data corroborates previous studies conducted in Nigeria [9,24,31,32], indicating a similarly high incidence of *Salmonella Typhi* in children in Nigeria. However, while most of these studies were conducted in clinically confirmed typhoid fever cases, our study was conducted by testing presumed healthy children with absence of fever. Our findings reveal that the burden of *Salmonella Typhi* cannot be ascertained by clinical cases alone. This is because healthy children can become carriers, spread and maintain *Salmonella Typhi* circulation in endemic environments. Therefore, to show the actual burden of *Salmonella Typhi* disease, there is a need for periodic community surveillance within and outside the hospital setting; this will further provide a prevalence statute of *Salmonella Typhi* disease in a healthy community, especially in endemic regions. In addition, screening these healthy carriers will help to control the spread of the infection to the public.

The Widal test, the major antigen test for detecting *Salmonella Typhi* in Nigeria, has some challenges. It is non-specific and is unable to differentiate active infection from passive or carrier infection. Previous studies on typhoid fever in Nigeria have utilized a non-specific Widal test technique to confirm the disease incidence [24,31,32]. Our study used a more robust ELISA assay for antigens and antibodies to *Salmonella Typhi* LPS. Thus, we confirmed evidence of active infection (39%) within the study cohort of enrolled children who harbored the *Salmonella Typhi* LPS antigen. Of these 47 children with positive antigens, seventeen (17) had the *Salmonella Typhi* LPS antigen and both antibodies (IgG and IgM) concurrently. Despite the absence of symptoms, we could distinguish an active infection from an inactive infection status based on the antigen or antibody status in the study population. This differs from the Widal test, the most common test for detecting *Salmonella Typhi* in endemic countries such as Nigeria. This distinction helps to confirm actual cases that require treatment, while the inactive cases, which are expected, confirm the endemicity of the infection. The concurrent occurrence of *Salmonella Typhi* LPS antigen and both antibodies further confirms the endemicity and prevalence of *Salmonella Typhi* in Nigeria. This agrees with a study conducted in Southern Vietnam that reported elevated LPS antibodies in both *Salmonella Typhi* confirmed cases and healthy controls, suggesting that background immunity to LPS could be high in typhoid fever endemic areas [22]. The endemicity of typhoid fever may contribute to the lack of visible symptoms in this group of children, even in the presence of high antigen titer levels reported. Furthermore, the high prevalence confirmed in this study is mainly within children above five years, which agrees with reports from other published studies of confirmed typhoid fever cases [33,34]. They report an increased fever duration among children over five years. This was not the case in this study population, although previous exposure may contribute to high seroprevalence in this age group.

Bacterial culture remains the gold standard for diagnosing typhoid fever. Surprisingly, fecal samples from both antigen and antibody-positive samples were negative for *Salmonella Typhi*, as the API kit could not confirm its presence in any of the samples. The exact reason for this is unclear, however, the possible presence of antibiotics in the system during sample collection may affect the detection of *Salmonella Typhi* in stool samples [35]. Indiscriminate drug use due to over-the-counter purchase of antibiotics is widespread in Nigeria [36]. Interestingly, all samples tested for the presence of *Salmonella Typhi* in the LPS antigen-positive samples were also negative by PCR and WGS. Negative conventional PCR for *Salmonella Typhi* O group gene does not necessarily imply the absence of *Salmonella Typhi*; it could be due to a low PCR detection threshold due to the human DNA background in the samples and PCR inhibitors [37]. It is also possible that, since the ELISA only detects the bacteria protein in the serum, both the culture and molecular assays detect the live organism or the nucleic materials that may be below the detection threshold. It is also possible that the organism has been cleared by the host immune system while the protein remains in the circulatory system for a while. However, the detection of *Salmonella Paratyphi A* in some samples by PCR might explain the negative PCR result in antigen-positive *Salmonella Typhi* samples. *Salmonella Paratyphi A* has been implicated in *Salmonella Typhi* ELISA cross-reactivity due to shared O12 LPS antigen [38].

Although the children were healthy and not tested for other infections, the subsequent metagenomics sequencing approach did not detect *Salmonella Typhi* in selected fecal antigen-positive samples selected with high O.D. values. However, other gut microbiotas, of which some are known pathogens, were detected and characterized. Detected reads for these bacterial species, such as *Klebsiella*, *Collinsella*, *Clostridium*, *Brachyspira*, *Blautia*, *Bifidobacterium*, *Anaerostipes*, and *Anaerobutyricum* were seen (Figure 4). Absence of *Salmonella* reads in fecal samples does not entirely exonerate healthy children as chronic carriers. They could harbor the pathogen in the gallbladder even when not seen in the blood or stool [39]. It is also possible that the low abundance of *Salmonella Typhi* compared to the detected bacteria in the stool samples is perhaps the reason for the absence of symptoms to confirm clinical cases in the enrolled children. This indicates a gap in diagnosing *Salmonella Typhi* carriers in healthy children with positive immunological markers. A study reported that positive immunological markers for typhoid fever had been associated with *Salmonella Typhi* carriage in confirmed cases of typhoid fever [23].

Our study has some limitations, including the use of a commercially available kit that may have yet to consider the endemicity of typhoid fever, and that may have influenced the data we generated. Secondly, comparison with similar research work was impossible due to the lack of published data using *Salmonella Typhi* LPS ELISA in this part of the world, as more *Vi* and *HlyE* antigen and Widal tests were reported. The lack of *Salmonella Typhi* LPS data in our study population makes this work a vital reference document for future research. Although the ELISA technique and the use of LPS for detecting *Salmonella Typhi* is rare, it can be optimized due to the relevance of LPS to microbial survival.

This study showed a high seroprevalence of typhoid fever in otherwise healthy children using a more robust ELISA assay compared to the commonly used Widal test. We also confirmed the need for proper sero-surveillance of *Salmonella Typhi* in endemic areas in order to ascertain the actual burden of typhoid fever in Nigeria. The systematic employment of different techniques in this study shows a disparity between serology, culture, and molecular processes in accurately confirming the prevalence of *Salmonella Typhi* in healthy children. This indicates that one technique alone may be insufficient for the surveillance of *Salmonella Typhi* in healthy children, especially in endemic areas [40]. Furthermore, we recommend more studies in other parts of the country to ensure adequate surveillance of *Salmonella Typhi* in healthy children in order to eliminate the disease. Moreover, introducing the typhoid conjugate vaccine will aid in eliminating the disease, as the typhoid vaccine is not currently included in the routine immunization scheme in Nigeria.

## 5. Conclusions

In conclusion, our study suggests a high seroprevalence of typhoid fever amongst healthy children who did not harbor the *Salmonella Typhi* pathogen by culture and molecular detection. We recommend that more studies be conducted to fully ascertain the detection threshold of the mentioned techniques and the burden of typhoid fever in healthy children in Nigeria and other endemic areas in order to achieve the eradication and elimination of the disease.

## Figures and Tables

**Figure 1 pathogens-12-00594-f001:**
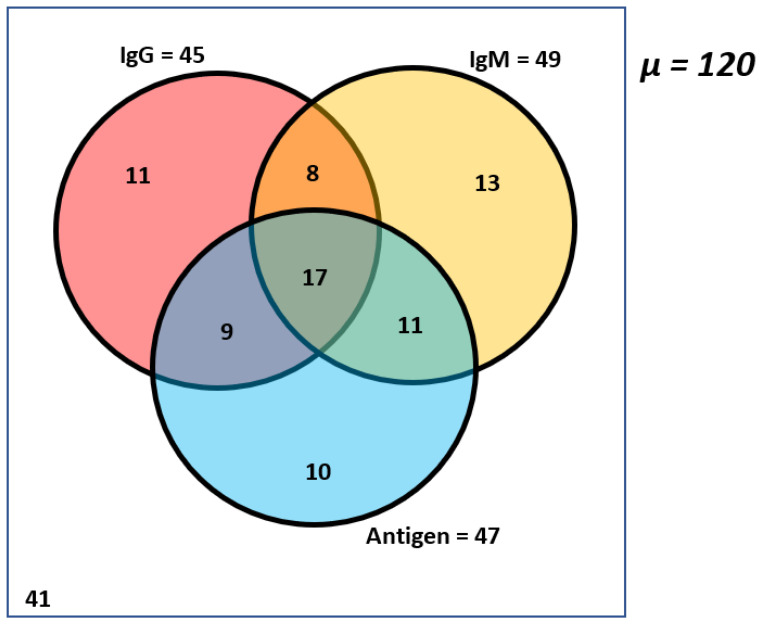
A Venn diagram showing the distribution of the antigen and the antibodies in the cohort of children who were positive for either or both or all biomarkers (*Salmonella Typhi* antibodies and antigen) *µ* = 120 means total samples analyzed.

**Figure 2 pathogens-12-00594-f002:**
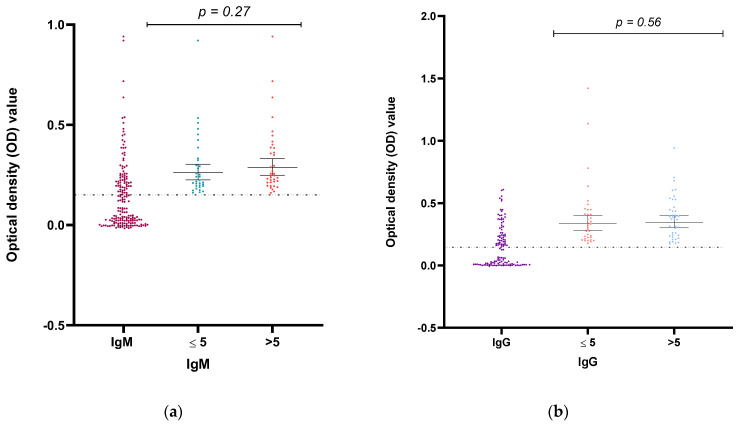
A scatter plot showing the prevalence of *S. Typhi* anti-LPS IgM (**a**) and IgG (**b**) in the study cohort based on the children’s O.D. distribution according to age groups for those who were positive. The thin line across the plot indicates the cutoff value (0.15). The *p*-value represents the comparison between ages ≤5 and >5.

**Figure 3 pathogens-12-00594-f003:**
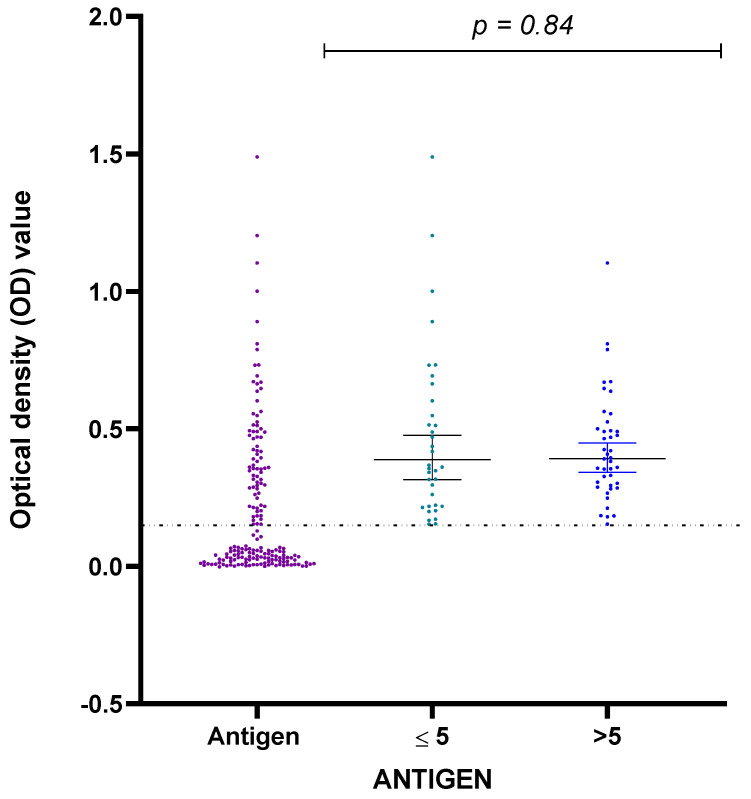
A scatter plot showing the prevalence of *Salmonella Typhi* LPS antigen in the study cohort of children based on the O.D. distribution according to age groups for those positive. The thin line across the plot indicates the cutoff value (0.15). The *p*-value represents the comparison between ages ≤5 and >5 years.

**Figure 4 pathogens-12-00594-f004:**
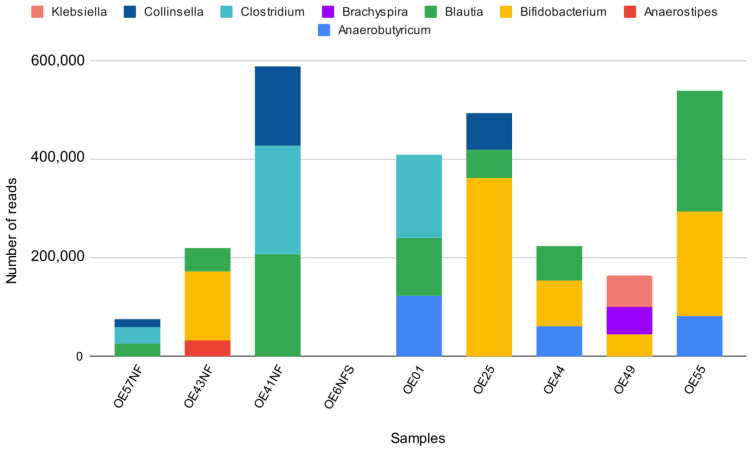
Detected bacteria genera in the stool of antigen-positive and negative samples. *Blautia* spp. and *Bifidobacteria* spp. are the most detected bacteria genera across all samples.

**Table 1 pathogens-12-00594-t001:** Primers sequences for O group and H (Flagella) antigen amplification.

Target	Forward	Reverse
O-group	Tyv-5′GAG GAA GGG AAA TGA AG C TTT T-3′ Prt-5′CTT GCT ATG GAA GAC ATA ACG AAC C-3′	5′-TAG CAA ACT GTC TCC CAC CAT AC-3′ 5′-CGT CTC CAT CAA AAG CTC CAT AGA-3′
H-antigen	H-5′ACT CAG GCT TCC CGT AAC GC-3′	Hd-5′GGC TAG TAT TGT CCT TAT CGG-3′ Ha-GAG GCC AGC ACC ATC AGT GC

**Table 2 pathogens-12-00594-t002:** Demographic profile of enrolled children.

	N	120 (%)
Sex (%)	Female	49 (40.8%)
	Male	71 (59.2%)
Age (years)	Mean ± S.D	6.1 ± 2.9
	95% Confidence interval	5.6–6.6
	Range	1–14
	Lower quartile	4
	Median	6
	Upper quartile	8
	≤5 years (%)	57 (47.5%)
	>5 years (%)	63 (52.5%)

**Table 3 pathogens-12-00594-t003:** Microbiology detection of *Salmonella Typhi*.

Test	Sample Size	The Outcome for *S. Typhi*
Salmonella/shigella agar culture	67	23 positives
API	23	None

API = Analytical profile index.

**Table 4 pathogens-12-00594-t004:** PCR detection of *Salmonella Typhi* and *Salmonella Paratyphi A*.

	PCR Target	Number Positive	Number Negative
*Salmonella Typhi* O-antigen	*tyv*	0	47
*Salmonella Typhi* H-antigen	*Hd*	0	47
*Salmonella Paratyphi* O-antigen	*prt*	6	41
*Salmonella Paratyphi* H-antigen	*Ha*	0	41

Somatic genes—*Tyv* and *Prt*; Flagellar genes—*Hd* and *Ha*.

## Data Availability

The sequence reads generated from this study can be found on NCBI under the BioProject accession number PRJNA929356.

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
