# Peer review of "The Prevalence of Undiagnosed Salmonella enterica Serovar Typhi in Healthy School-Aged Children in Osun State, Nigeria"

_pathogens, 2023, doi:10.3390/pathogens12040594_

Round 1
Reviewer 1 Report
- Why two different ranges of wavelength were selected for Delta O.D. (Optical Density) (O.D. at 630nm subtracted from O.D. at 450nm) 145 for each sample, which was blanked by subtracting the O.D. value of the blank as described in Section 2.10?
- How the identified strains are different from previously reported S. typhi
- What are the reasons why people in endemic areas are more susceptible to typhoid fever?
- Genome sequencing information is missing and needs to be included.
- PCR methodology and gel picture is need to incorporate.
Reviewer 2 Report
Investigation of prevalences of undiagnosed pathogen carriage such as Salmonella typhi is an important pillar of disease prevention and a prerequisite for effective pathogen eradication measures. However, Uwanibe et al. provide with their manuscript hardly data which would allow insights into Salmonella typhi prevalence in the studied children’s cohort. For serology data it should be discussed and literature referenced which S. typhi LPS- and antigen epitopes are targeted by antibody isotypes, i.e. it is not enough to just name the ELISA kit used. Positive and negative controls should be used in assays in order to exclude cross reactivity or artefacts. In the same line qPCR and bacterial culture data must be presented. In the absence of showing that the methods work for controls, no conclusions can be drawn on negative results at all. Finally, the use of metagenome sequencing is not sensitive enough to detect low level S. typhi carriage. The presented figure 4 indicated methodological deficiencies with regard to variability and complexity. In general, published data on S. typhi prevalences in other regions, successfully used detection methods for S. typhi and prevention measures are not sufficiently discussed and cited.
Reviewer 3 Report
The authors describe the prevalence of undiagnosed Salmonella Typhi in healthy School-Aged children in Osun state, Nigeria which is an important and interesting subject since typhoid fever is a serious public health problem and symptomatic carriers are a reservoir of typhoid fever. Although the subject of the work is interesting the research results are a bit outdated (December 2019- February 2020).
What is more the manuscript need some major revision to be accepted:
- Title: “Salmonella typhi” it should be “Salmonella Typhi” or Salmonella enterica serovar Typhi this serious error should be corrected throughout the manuscript
- I suggest the Authors to shorten the introducion.
- Please remove hyperlink from literature references.
- Some subheadings have dots at the end, others don't. Please standardize according to the requirements of the Journal
- Line 118 à °C
- Line 125 à please use correct nomenclature throughout the manuscript
- Line 158 à p-values
- Table 2 - to make the table more complete, I suggest adding the median, lower quartile and upper quartile
- Line 173 - please use either abbreviations or full species names throughout the text
- Line 318 à . Our
- I suggests the Authors to add a conclusion section at the end of the manuscript
Round 2
Reviewer 1 Report
All the comments raised have been addressed and are satisfactory.
Reviewer 2 Report
Raised reviewer issues have been improved, therefore the manuscript in the present form is acceptable for publication.
Reviewer 3 Report
Thank you for significant improvements to this manuscript.